# acCRISPR: an activity-correction method for improving the accuracy of CRISPR screens

Adithya Ramesh[1,6], Varun Trivedi [1,6], Sangcheon Lee[1,6], Aida Tafrishi[1], Cory Schwartz[1,5], Amirsadra Mohseni [2], Mengwan Li[1], Stefano Lonardi [2,3] & Ian Wheeldon [1,3,4 ✉]

High throughput CRISPR screens are revolutionizing the way scientists unravel the genetic underpinnings of engineered and evolved phenotypes. One of the critical challenges in accurately assessing screening outcomes is accounting for the variability in sgRNA cutting efficiency. Poorly active guides targeting genes essential to screening conditions obscure the growth defects that are expected from disrupting them. Here, we develop acCRISPR, an end-to-end pipeline that identifies essential genes in pooled CRISPR screens using sgRNA read counts obtained from next-generation sequencing. acCRISPR uses experimentally determined cutting efficiencies for each guide in the library to provide an activity correction to the screening outcomes via calculation of an optimization metric, thus determining the fitness effect of disrupted genes. CRISPR-Cas9 and -Cas12a screens were carried out in the non-conventional oleaginous yeast *Yarrowia lipolytica* and acCRISPR was used to determine a high-confidence set of essential genes for growth under glucose, a common carbon source used for the industrial production of oleochemicals. acCRISPR was also used in screens quantifying relative cellular fitness under high salt conditions to identify genes that were related to salt tolerance. Collectively, this work presents an experimental-computational framework for CRISPR-based functional genomics studies that may be expanded to other non-conventional organisms of interest.

[1] Department of Chemical and Environmental Engineering, University of California, Riverside, CA 92521, USA. [2] Department of Computer Science and Engineering, University of California, Riverside, CA 92521, USA. [3] Integrative Institute for Genome Biology, University of California, Riverside, CA 92521, USA. [4] Center for Industrial Biotechnology, University of California, Riverside, CA 92521, USA. [5] Present address: iBio Inc., San Diego, CA, USA. [6] These authors contributed equally: Adithya Ramesh, Varun Trivedi, Sangcheon Lee. ✉email: wheeldon@ucr.edu

Functional genetic screening with pooled libraries of CRISPR guides has been successful in discovering gene function, identifying essential genes, and evolving new phenotypes[1–3]. These screens work by inducing mutations across the genome to disrupt gene function. Genome-wide transcriptional regulation is also possible when a catalytically deactivated Cas endonuclease (typically, Cas9 or Cas12a) fused to an activation or repression domain is targeted to promoters[4,5]. For these screens to be effective, the library should contain one or more active guide RNAs for each targeted gene. Creating such libraries is challenging due to imperfect design algorithms and an incomplete understanding of how Cas endonucleases function across different species. Further confounding guide design is the blocking effect of chromatin structure on guide RNA targeted Cas9 endonuclease[6,7]. As a result of this imperfect design, CRISPR screens are conducted with pooled libraries of guide RNAs that have a broad range of activity[8,9]. High activity guides can assign phenotypic changes to genome edits with high confidence, while inactive and low activity guides can obscure gene hits by producing false negatives. Computational and experimental methods that can quantify the activity of each guide in a library and account for the variance in activity are needed to correct screening outcomes, accurately identify genotype-phenotype relationships, and call essential genes with high confidence.

A common CRISPR library design strategy is to include many guides targeting each gene or promoter. This strategy helps ensure that every gene is targeted by an active guide, but doing so increases the analytical complexity in assessing outcomes. Current analysis methods use a Bayesian framework to infer guide activity from screens obtained across several experimental conditions; guide RNAs that elicit a fitness effect under several different conditions are indicative of high activity[10,11]. Reliable measurements of guide activity can also be generated directly from screening experiments. In the yeast species that we have studied[12], this can be achieved by disrupting the primary DNA repair mechanism (typically, non-homologous end-joining or NHEJ) and using negative growth selections to quantify the activity of each guide, resulting in activity profiles across the genome. Guide activity data, whether computationally or experimentally produced, is used to identify and account for inactive and low activity guides, leading to improved hit calling and screen accuracy. Here we show that, given experimental guide activity measurements from a single screen, significant hits can be identified using average $\log_2$-fold change, thereby eliminating the need to process multiple screens and perform probabilistic modeling of the data.

In this work, we develop an activity-correction CRISPR screen analysis method—acCRISPR—that optimizes library activity to generate accurate screening outcomes. Using guide RNA abundance data from sample and control screens along with information on the activity of each guide, acCRISPR computes a fitness score for every targeted gene and identifies genes essential to the screening condition. We demonstrate the utility of acCRISPR by analyzing CRISPR-Cas9 and -Cas12a screens in negative selection experiments in the oleaginous yeast *Yarrowia lipolytica*. We focus on this yeast because it has the ability to synthesize and accumulate lipids, and for its success as a host for oleochemical biosynthesis[13–15]. Using previously derived guide activity profiles of *Yarrowia* genome-wide Cas9 and −12a libraries (see ref. [16]), along with new growth screens, we use acCRISPR to identify essential genes and call hits in high salt tolerance screens. We independently validate acCRISPR predictions by measuring growth of individual disruptions of a subset of essential genes and tolerance genes in conditions akin to those of the original genome-wide screens. We also evaluate the performance of acCRISPR with computational predictions of guide activity rather than experimentally determined values. Essential gene analysis and functional genetic screening will help toward developing a better understanding of *Yarrowia*'s genetics, and acCRISPR analysis of the screens conducted in this work enables this.

## Results

**acCRISPR optimizes sgRNA library activity and coverage.** acCRISPR uses raw read counts of guide RNAs from functional screens as inputs and computes cell fitness effects, guide RNA activity profiles, and calls essential genes. To demonstrate this analysis pipeline, we conducted CRISPR-Cas9 and -Cas12a genome-wide screens in the PO1f strain of *Y. lipolytica*. The pooled guide libraries contain single guide RNAs (sgRNAs) that target more than 98.5% of the protein-coding sequences with 6- and 8-fold coverage for Cas9 and Cas12a, respectively. Guide activity in these libraries was previously reported[9,16]; a cutting score (CS), defined as the $-\log_2$ ratio of normalized read counts obtained in PO1f Cas9/12a *ΔKU70* to counts in the control strain, was determined for each guide (Fig. 1a). The disruption of *KU70* disables NHEJ DNA repair[17], creating a link between guide abundance in a negative selection growth screen and guide activity. In the absence of the dominant DNA repair mechanism, a double-stranded break causes cell death or significant impairment in growth; sgRNAs with high activity are lost from the cell population with higher frequency than those with lower activity, thus linking CS to guide activity. The fitness screen inputs for acCRISPR were generated using PO1f as the control strain and PO1f Cas9 or Cas12a as the sample. Screens were conducted in synthetic defined media with glucose as the sole carbon source. An Illumina sequencing instrument was used to generate sgRNA read counts after four days of culture. These data were used to generate a fitness score (FS) profile, defined as the $\log_2$ ratio between the normalized counts in the Cas9/Cas12a expressing strain and the control. Raw guide RNA counts for Cas9 and Cas12a screens are provided in Supplementary Data 1 and 2.

The first analytical step of acCRISPR is to convert raw guide abundance values into CS and FS profiles (Fig. 1b, Supplementary Data 3). First, an FS is computed for each gene as the average $\log_2$-fold change of all guides targeting that gene, both active and inactive. Then, the FS value for each gene is recalculated after excluding sgRNAs with a CS below a given CS threshold (i.e., a minimum value of CS for an sgRNA to be included in the analysis, $T$). As guides with low CS are removed, the library coverage is reduced along with the statistical power that multiple guides provide. To capture this effect, we compute the ac-coefficient as the product of the CS threshold ($T$) and the average number of guides per gene, for a range of $T$ values. The maximum peak for the ac-coefficient indicates the CS threshold where the library activity is maximized. The corrected FS profile generated for the threshold corresponding to the peak is used to identify essential gene hits; p-values for every gene in the dataset are determined by comparing the FS of a gene to a null distribution that represents the fitness of non-essential genes (see "Methods" for more details).

**acCRISPR accurately calls essential genes.** We evaluated the performance of acCRISPR against other established approaches that classify essential genes using read counts or $\log_2$-fold changes from CRISPR screens as input, namely JACKS[10], MAGeCK-MLE[11], and CRISPhieRmix[18]. These methods have been validated against a gold standard set of essential genes in mammalian cells and were used here to compute fitness effects and call essential genes in *Yarrowia*. The comparison of acCRISPR to the other methods on our Cas9 screens is shown in Fig. 2. Similar analyses

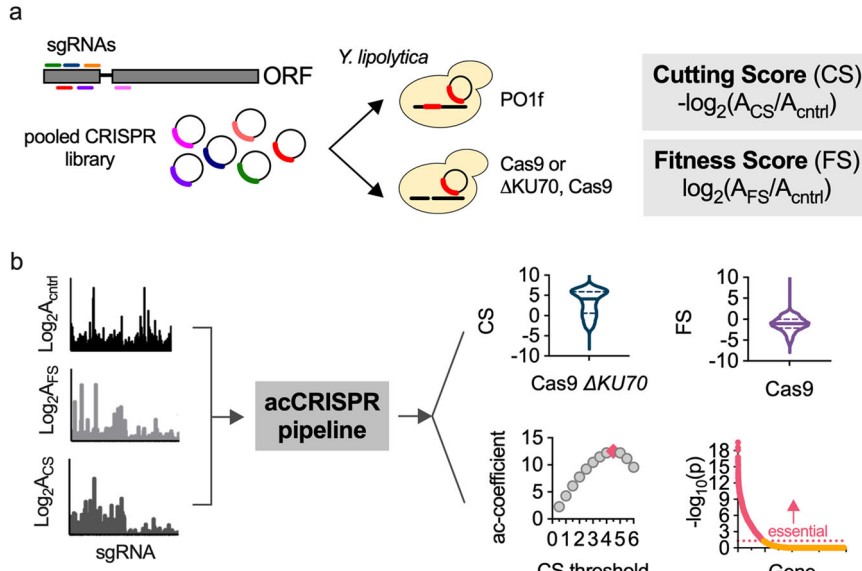

**Fig. 1 acCRISPR analysis of CRISPR-Cas screens. a** Growth screens in *Y. lipolytica* were conducted with pooled libraries of single guide RNAs (sgRNAs) (6- and 8-fold coverage of >98.5% of CDSs, for Cas9 and Cas12a respectively). A guide's cutting score (CS) is equal to the $-\log_2$ fold-change of normalized guide abundance in PO1f Cas9/12a $\Delta KU70$ to the control strain. Fitness scores (FS) are similarly defined, but with the PO1f Cas9/12a strain as the sample. **b** acCRISPR takes normalized sgRNA read counts from the control, CS, and FS strains and computes a series of outputs: CS per guide, FS per gene, the ac-coefficient (the product of $CS_{threshold}$ and library coverage), and *p*-value per gene from significance testing against a non-essential gene population at the maximum ac-coefficient. The data sets shown here are from Cas9 screens in *Y. lipolytica* PO1f. Screens were conducted at 30 °C with glucose as the sole carbon source. Genes with an essentiality *p*-value < 0.05 were classified as essential.

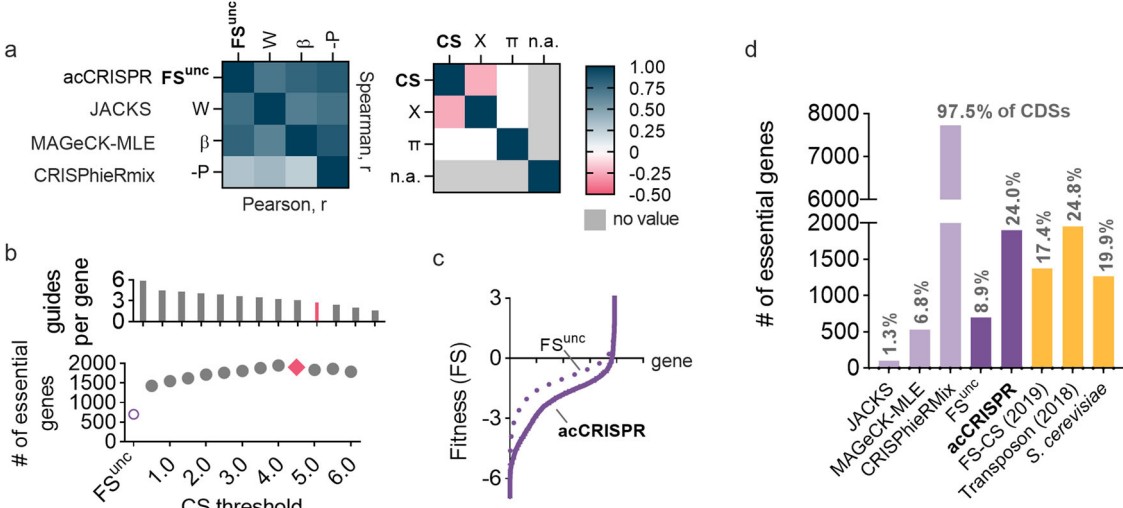

**Fig. 2 acCRISPR analysis of CRISPR-Cas9 screens defines a high confidence set of essential genes. a** Heat maps showing Pearson (below diagonal) and Spearman (above diagonal) correlation coefficients for comparison of gene fitness effects (uncorrected FS (FS$^{unc}$), W, β, and -P; left) and sgRNA cutting efficiencies (CS, X, and π; right) from acCRISPR and three established essential gene identification algorithms, JACKS, MAGeCK-MLE and CRISPhieRmix. 'n.a.' denotes that sgRNA cutting efficiency values for CRISPhieRmix are not available. **b** The average number of sgRNAs per gene (top) and the number of essential genes predicted (bottom) with increasing CS threshold as well as for uncorrected FS. The data points colored in pink are the guides per gene and the number of essential genes determined at the maximum ac-coefficient. **c** Fitness scores of genes with (solid line) and without (dashed line) acCRISPR processing with a CS threshold (*T*) of 4.5. **d** The number of essential genes identified by JACKS, MAGeCK-MLE, CRISPhieRmix, FS$^{unc}$, and acCRISPR are compared to previously reported essential gene sets for *Yarrowia* (FS-CS[9] and transposon analysis[19]) and *S. cerevisiae*[20]. Values at the top of each bar indicate the percentage of the total number of genes identified as essential by the respective method.

of the CRISPR-Cas12a screens are shown in Supplementary Fig. 1.

Output values for the fitness effect of genes in *Yarrowia* from acCRISPR, JACKS, and MAGeCK-MLE (FS uncorrected (FS$^{unc}$), W, and β) are in good agreement. The pairwise Pearson and Spearman r-values are 0.65 or greater (Fig. 2a). CRISPhieRmix

was less successful at capturing raw fitness effects from the *Yarrowia* screen (Pearson r < 0.37) and the majority of genes were identified as essential. JACKS and MAGeCK-MLE also output guide activity predictions (X and π); these values did not correlate well with the acCRISPR analysis of the CS profiles, which were directly obtained from the screening experiment.

a

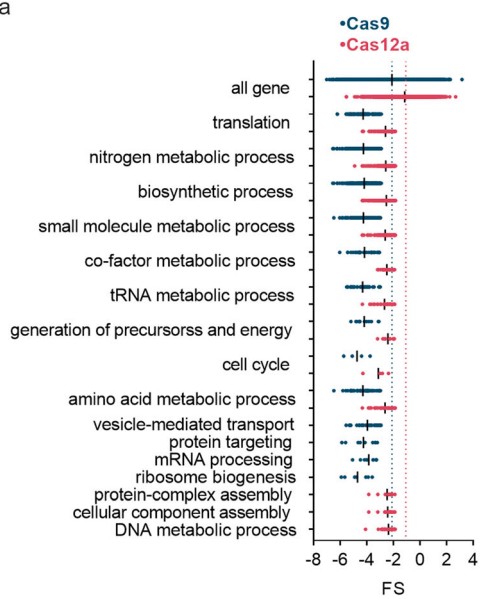

b

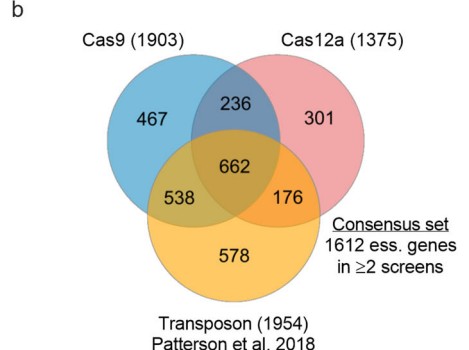

**Fig. 3 Defining a set of consensus essential genes in *Y. lipolytica*.** **a** Enriched GO-Slim biological process terms for Cas9 and Cas12a essential gene sets and FS distribution of essential genes associated with each GO-Slim term. Enriched terms were determined using a hypergeometric test (FDR-corrected, *p* < 0.05). The FS values for each GO-Slim term were found to be significantly lower than those of all genes by one-tailed unpaired *t*-test (*p* < 0.0001). Blue and red dotted lines indicate the mean FS of all genes for Cas9 and Cas12a datasets respectively. **b** Venn diagram of the essential genes identified from CRISPR-Cas9, CRISPR-Cas12a, and transposon screening, and their overlap. The consensus set of essential genes, comprising genes common to at least two of the three screens, contains 1612 unique genes.

We next applied CS correction to the Cas9 screening data. The ac-coefficient curve for the Cas9 screen for each choice of the CS threshold *T* is shown in Fig. 1b. The number of essential genes and the average number of guides per gene for the same values of the threshold *T* are shown in Fig. 2b. As *T* increased from 0.5 to 4.0, the number of genes classified as essential also increased, an effect likely caused by removing false negatives resulting from poor activity sgRNAs targeting essential genes. The optimum library activity, indicated by the peak of the ac-coefficient, occurred at threshold *T* = 4.5 with an average coverage of 2.78 guides per gene. The peak for the ac-coefficient in the CRISPR-Cas12a library indicated the optimal CS threshold of *T* = 1.5, with an average coverage of 2.97 guides per gene (Supplementary Fig. 1).

The optimized acCRISPR analysis of the Cas9 screen identified 1903 essential genes (see Supplementary Data 4), a number similar to the 1954 essential genes reported for a transposon-

based screen[19]. Without the activity correction, only 702 genes could be classified as essential, a value significantly below what was expected; based on the analysis of other yeast species ~15% to ~30% of protein-coding genes are expected to be essential (e.g., 19.9% for *S. cerevisiae* and 26.1% for *S. pombe*[20,21]). The Cas12a screens conducted here identified 1375 genes as essential (Supplementary Data 4) when the acCRISPR pipeline was used, and only 335 when all sgRNAs (both active and inactive) were included in the analysis. JACKS and MAGeCK-MLE also underpredicted the number of essential genes in the Cas9 and Cas12a screens (JACKS, 102 and 0; MAGeCK-MLE, 535 and 1218), while CRISPhieRmix classified nearly all genes as essential (7724 and 7538).

**CRISPR-Cas9 and -Cas12a screens help define a consensus set of essential genes.** The acCRISPR analysis of the Cas9 and -12a screens provides the opportunity to define a consensus set of essential genes for *Yarrowia* growth on glucose. First, we validated the essential gene set via a Gene Ontology (GO) enrichment analysis[22,23], with the expectation that functional terms known to be essential would be enriched (FDR-corrected *p* < 0.05; see Supplementary Data 5 and 6 for all GO and GO-Slim terms pertaining to molecular function (MF), biological process (BP) and cellular component (CC)). As expected, genes involved in transcription, translation, cell cycle regulation, cofactor metabolism, and tRNA metabolic processes showed significantly lower FS values (*t*-test, *p* < 0.05) compared to the average FS of all genes in both the Cas9 and Cas12a screens. The FS values of genes in these functional groups along with other enriched GO-Slim terms are shown in Fig. 3a.

A previously published transposon-based screen identified 1954 essential genes[19]. Experimental conditions (2% glucose in SD-Leu media) were consistent with the Cas9 and Cas12a experiments conducted here, thus providing a large data set from which we can identify a consensus set of essential genes. One thousand six hundred and twelve genes were common to at least two of the three different screens (Fig. 3b and Supplementary Data 7). Enriched GO-Slim terms in this set were consistent with those expected for essential genes and we consider these genes as the consensus set for *Yarrowia* growth on glucose (see Supplementary Data 8). To verify the essentiality of genes in the consensus set, we tested 15 essential genes from this set and 5 non-essential genes (i.e., genes non-essential in all 3 screens) using the CRISPR-Cas9 system and measured their abundance in glucose after 2 days of growth (Supplementary Fig. 2; see Methods for details on the experimental procedure). Of the 15 essential genes tested, 12 were called as essential in all three screens, while 3 others were called as essential in the Cas9 and Cas12a screens, but not in the transposon screen. As expected, cells containing essential gene knockouts showed no or minimum growth throughout the validation experiment, whereas disruptions of non-essential genes exhibited substantial growth over the same time period. One-tailed t-test indicates that the growth of non-essential gene knockouts is significantly higher (*p* < 0.0001) than that of the essential gene knockouts. The essential genes identified in the consensus set were also compared to known essential genes in *S. cerevisiae* and *S. pombe*. Of these, 824 genes were identified to have homologs in *S. cerevisiae*, of which 54.6% were found to be essential in both species. Seven hundred and eighty-two genes had homologs in *S. pombe* and 60.9% of those were found to be commonly essential between both species (Supplementary Fig. 3).

**acCRISPR can use sgRNA activity predictions as an alternative to CS.** We recognize that generating experimental CS profiles is not always feasible (for example, in organisms for which it is not

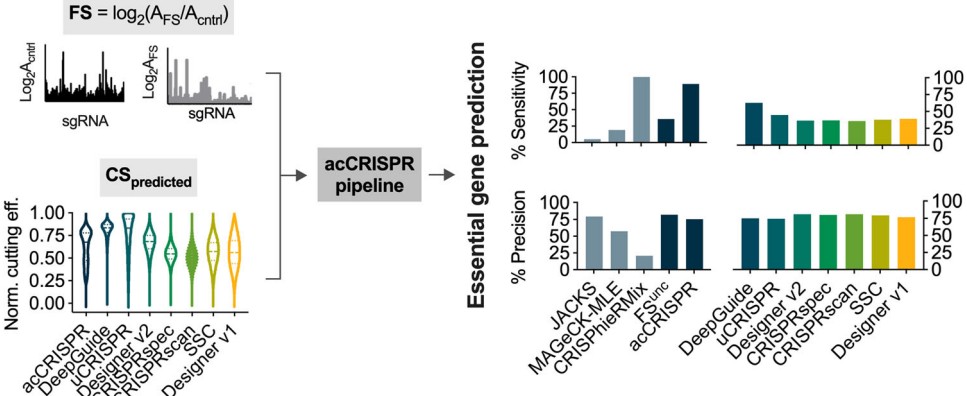

**Fig. 4 Performance of acCRISPR using predicted sgRNA activity profiles in *Y. lipolytica*.** Raw sgRNA counts from control and treatment strains used for fitness score calculations were provided as input to acCRISPR along with sgRNA activity scores from a range of guide activity prediction tools (DeepGuide[16], uCRISPR[24], Designer v2[26], CRISPRspec[29], CRISPRscan[28], Spacer Scoring for CRISPR (SSC)[27], and Designer v1[25] left). The violin plot shows the distribution of min-max normalized CS (denoted by 'acCRISPR') and sgRNA activity scores from each prediction tool. Dashed lines represent the median of the normalized score and the dotted lines represent the first and third quartiles. Essential genes were identified using predicted sgRNA efficiency scores from each tool after first determining the maximum ac-coefficient. The % sensitivity and % precision in identifying genes from the consensus set are shown (right). Bars indicate the values of these two metrics for each prediction tool as well as for JACKS, MAGeCK-MLE, CRISPhieRmix, uncorrected FS (FS$^{unc}$), and acCRISPR.

possible to have NHEJ-deficient screens or in cases where a double stranded break is likely to be repaired by homology directly using a second allele as a template). Thus, we sought to test the performance of acCRISPR using computationally predicted sgRNA activity scores in *Yarrowia*. Among the large set of guide activity prediction tools available for Cas9, we selected DeepGuide[16], uCRISPR[24], Designer v1[25], Designer v2[26], SSC[27], CRISPRscan[28], and CRISPRspec[29] (Fig. 4 and Supplementary Data 9). For Cas12a, only a few prediction algorithms have been developed, for example, DeepGuide[16] and DeepCpf1[30], which have been shown to predict sgRNA activities in *Yarrowia* with reasonable accuracy (Supplementary Fig. 4 and Supplementary Data 10). Using the predicted activity scores, we implemented acCRISPR to compute the maximum ac-coefficient (Supplementary Table 1) and determined a set of predicted essential genes. The consensus set identified in Fig. 3 served as a reference to evaluate the success of each prediction method. Of all prediction methods, DeepGuide was found to have the highest sensitivity for both Cas9 (62.8%) and Cas12a (51.7%) datasets (where sensitivity is the percentage of the consensus set that is captured by the predicted set). The higher performance of DeepGuide is likely a consequence of its training set, that is the *Yarrowia* CS profiles generated in our screens. Other methods captured a smaller fraction of the consensus set, with sensitivity ranging from 26.0 to 44.9%. While the predicted guide activities were not successful at capturing the full set of essential genes in *Yarrowia*, those that were identified were called with high confidence; each of the tested methods maintained precision rates above ~75% (where precision is the number of predicted essential genes overlapping with the consensus set divided by the total number of essential genes predicted).

In addition to evaluating the success of different guide prediction algorithms, we determined sensitivity and precision metrics for Cas9 and Cas12a screens using acCRISPR, JACKS, MAGeCK-MLE, CRISPhieRmix, and uncorrected FS profiles, with CS as an input (Fig. 4 and Supplementary Fig. 4). acCRISPR analysis of the Cas9 screen captured nearly all of the consensus set (sensitivity of 89.1%) with high precision (75.5%). Except for CRISPhieRmix, the other methods failed to capture the majority of the consensus set. CRISPhieRmix classified nearly all *Yarrowia* genes as essential, thus capturing nearly 100% of the consensus

set but with low precision (20.8%). Results of a similar analysis with the Cas12a screen are reported in Supplementary Fig. 4; the Cas12a screen captured 66.7% of the consensus set with 78.1% precision.

**acCRISPR identifies biologically insightful hits related to salt tolerance.** To further demonstrate the utility of acCRISPR, we conducted high salt tolerance screens from which we identified genetic hits that produced significant effects on cell fitness. Tolerance to high salinity is an industrially beneficial trait that can reduce costs associated with process sterilization and enable growth in lower-cost water sources (e.g., seawater or wastewater)[31]. The CRISPR-Cas9 strain was grown in the presence and absence of two different levels of salt concentration ([NaCl] of 0.75 and 1.5 M) and acCRISPR was used to identify significant hits for each salt stress condition. As a control, the Cas9-containing strain was grown under standard growth conditions (no added NaCl). In place of FS, these screens defined a tolerance score (TS), which is equal to the $\log_2$ ratio of sgRNA abundance under the stress condition (i.e., in the presence of salt) to that grown under control conditions (Fig. 5a). A low TS indicated that gene disruption conferred a growth disadvantage under the applied stress (see Supplementary Fig. 5 for corrected TS profiles in tolerance screens conducted at 0.75 M and 1.5 M NaCl).

acCRISPR analysis of the salt tolerance screens (Supplementary Fig. 6) identified 721 and 884 gene hits in 0.75 M and 1.5 M NaCl respectively (Supplementary Data 11). The two screening conditions were found to share 344 significant genes in common (Fig. 5b). Similar to the essential gene screening outcomes, we sought to validate a subset of the gene hits (see Methods for experimental details). The validation set included four genes: YALI1_E24201g (TS$_{1.5M\ NaCl}$ = −4.5), YALI1_E23961g (TS$_{1.5M\ NaCl}$ = −4.2), YALI1_F12478g (TS$_{1.5M\ NaCl}$ = −4.9), and YALI1_A07277g (TS$_{1.5M\ NaCl}$ = −4.7; significant only in 1.5 M NaCl). YALI1_E24201g and YALI1_E23961g were selected for validation because homologs of these genes are known to affect salt tolerance in other species. The GO-term of YALI1_E24201g suggests this gene encodes for 4-coumarate-CoA ligase, which has been shown to enhance abiotic stress tolerance, including salt

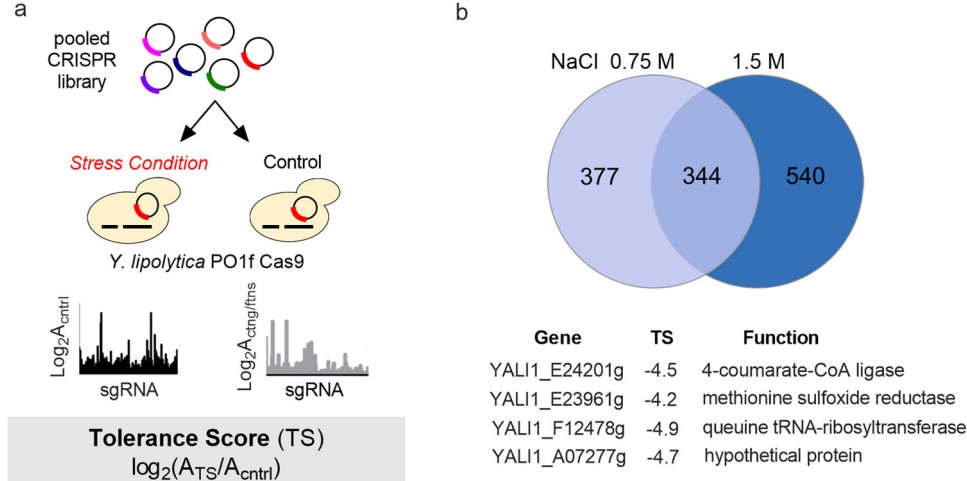

**Fig. 5 acCRISPR analysis of salt tolerance screens. a** Schematic of the CRISPR-Cas9 stress tolerance screens in *Yarrowia*. Analogous to fitness score (FS), the tolerance score (TS) is used to define the effect of each guide on cell growth under a stress condition. TS is equal to the $\log_2$-fold change of sgRNA abundance in the treatment to the control, where the control is a Cas9-expressing strain grown under standard culture conditions. **b** Outcomes of high salt tolerance screens. Venn diagram (top) shows the overlap of gene hits identified in the salt (0.75 M and 1.5 M NaCl) screens. Selected hits are shown (bottom), including the gene ID, the TS value from the 1.5 M NaCl condition, and putative gene function.

tolerance, in various plant species[32–34]. YALI1_E23961g is homologous to methionine sulfoxide reductase (*MXR1*) in *S. cerevisiae* and has been shown to improve resistance to oxidative stress[35]. The other two gene hits selected for validation, YALI1_F12478g (a queuine tRNA-ribosyltransferase) and YALI1_A07277g (a hypothetical protein), have no known connection to stress tolerance. In all four cases, gene disruption in individual experiments that mimicked the screening conditions resulted in significantly lower ($p < 0.01$) growth than the disruption of a gene with a higher TS value that was not called as significant by acCRISPR, thus validating the called hits (Supplementary Fig. 7).

Overall, the results reported here support the validity of our acCRISPR analysis in identifying novel gene hits related to salt stress tolerance; the full list of hits will enable us to identify new cellular functions related to stress tolerance as well as identify mutational targets for engineering new strains with increased tolerance.

## Discussion

A central challenge in analyzing CRISPR screens is deconvoluting the effect of poorly active guides from guides that create genome edits and elicit fitness effects. One approach to solving this challenge is to interrogate each edit in an arrayed format. The physical separation of different genetic perturbations throughout the screen also makes this approach more easily combined with -omics based profiling for further characterization of mutants. However, this requires extensive laboratory automation to achieve the throughputs that are accessible to pooled screens, where one can test the effect of all library mutants in a single culture. On the other hand, pooled screens lack distinct separation between mutants and thus rely on next generation sequencing methods to quantify the effect of genetic perturbations on cell fitness. Thus, resolving the effect of non performing guides becomes ever more important in this context. acCRISPR addresses this issue in pooled screens by optimizing the screen's ac-coefficient, a parameter that balances the trade-off between guide activity and coverage to maximize the performance of the library. In contrast to existing methods that infer sgRNA activity by modeling multiple screening conditions, acCRISPR uses an

experimentally derived measure of guide activity obtained from an additional treatment sample in which DNA repair by NHEJ is disrupted. This additional data enabled acCRISPR to outperform other approaches in determining an accurate set of essential genes.

acCRISPR was developed and validated using CRISPR-Cas9 and -Cas12a screening data to define essential genes in the oleaginous yeast *Y. lipolytica*. The other methods tested here, JACKS, MAGeCK-MLE, and CRISPhieRmix, are most commonly used to analyze the outcomes of mammalian cell CRISPR screens, and were found to be incompatible with our *Yarrowia* data; only a small percentage or all genes were identified as essential. This incompatibility is likely because the overlap between the fitness effect profiles of the non-targeting controls and the active sgRNA population is greater in mammalian cells compared to *Yarrowia* (Supplementary Fig. 8 and see refs. [18,36]). CRISPhieRmix, which uses the non-targeting population to form the null distribution, greatly overestimates the number of essential genes in *Yarrowia*, classifying nearly all genes as essential. The relative fitness effects that targeting and non-targeting sgRNAs have may also be harder to resolve in mammalian cells due to alternative splicing, polyploidy, and redundant gene function. acCRISPR, on the other hand, uses sgRNA targeting non-essential genes to construct the null model, thereby making it more adaptive to the *Yarrowia* dataset, and potentially more adaptable to other datasets.

While acCRISPR's use of an experimentally derived CS dataset is empowering, it also increases the technical difficulty of the experiments and is not necessarily accessible in all organisms (e.g., activity profiles across mammalian cell genomes and the genomes of other species have not yet been defined). We also recognize that alternate repair mechanisms could mask CRISPR Cas9/12a cutting. For example, we have previously observed error-prone microhomology mediated end-joining (MMEJ) DNA repair in *Yarrowia*[17]. sgRNA that produce such cases will likely result in negative CS and FS values, indicating that despite poor guide activity, gene editing still occurred at a rate sufficient to affect cell fitness. Analysis of the CS and FS values per guide reveal that only 1.2% and 2.1% of guides from the Cas9 and Cas12a libraries respectively fit this pattern (see Supplementary Data 3). The primary feature of acCRISPR is to remove guides with low CS, as such the majority of cases where an alternative

repair mechanism was active will likely be removed from the final analysis.

The ability to use predicted sgRNA activities in place of experimental activity scores may help address the limitation of requiring an experimental dataset. acCRISPR analysis with predicted activity resulted in high precision but modest sensitivity, thereby capturing a small portion of the essential genes but with high confidence (Fig. 4). While prediction methods have proven effective at designing active CRISPR sgRNAs, predictive power is still limited to the organism from which the training data was generated[8,16,37]. As better guide design algorithms are developed, we anticipate an improvement in acCRISPR performance in resolving essential genes when using predicted guide activities in place of experimentally derived CS distributions.

acCRISPR analysis of the screens conducted here represents a meaningful step toward understanding *Yarrowia* genetics. Thus far, there have only been a few attempts at classifying essential genes[9,19]. We use the CRISPR-Cas9 and -Cas12a screens conducted here along with the outcomes of a transposon screen conducted under similar conditions (see ref. [19]) to define a consensus set of essential genes for growth on glucose. This set contains 1612 genes that were classified as essential in at least two of the three independent screens, a subset of which were independently validated (Fig. 3 and Supplementary Fig. 2). While a considerable number of essential genes were called by 2 or 3 of the different technologies, a number of genes were unique to each, likely due to mechanistic differences between the mutagenesis strategies. For example, transposon-based screens have sequence biases for insertions and are known to miss shorter genes[38,39]; the more restrictive PAM of Cas12a leads to lower genome-wide coverage; Cas9 has been shown to have higher rates of off-target effects, which could lead to false predictions; and specific to our experiments, the Cas12a library contains more inactive and low activity guides, thus reducing the number of genes targeted by highly active sgRNAs. Defining a consensus set mitigates these differences as well as other potential issues with functional genomic screens (e.g., plasmid instability) and leads to calling a high confidence set of essential genes—that is, those that were called in more than one screen. GO term enrichment analysis suggests that genes in the consensus set have functions expected to be essential (e.g., genes related to transcription, translation, and cell cycle among others; Supplementary Data 8), while those unique to each method have no enriched functions (Supplementary Data 12).

With respect to the high salt tolerance screens, acCRISPR analysis also helps to advance our understanding of *Yarrowia* genetics by identifying high confidence hits with significantly decreased cell fitness, a subset of which were independently validated. This information promises to guide future strain engineering seeking to improve production host tolerance to harsh environmental conditions.

acCRISPR is an end-to-end pipeline for the analysis of pooled CRISPR screens. It takes a hybrid approach that combines experimental and computational methods to determine the activity of each guide in a pooled CRISPR screen and uses this information to correct screening outcomes based on guide activity. We use this pipeline to generate new knowledge on the genetics of *Y. lipolytica*, including the identification of a consensus set of essential genes for growth on glucose and calling loss of fitness hits for growth under high salt conditions. While this work focuses on analyzing screens conducted in *Y. lipolytica*, the same experimental-computational workflow can be readily applied to other organisms in which accurate computational prediction or genome-wide functional screens can be used to estimate sgRNA activities.

## Methods

**acCRISPR framework**. acCRISPR performs essential gene identification by calculating two scores for each sgRNA, namely the *cutting score* (CS) and the *fitness score* (FS). CS and FS are the $\log_2$-fold change of sgRNA abundance in the appropriate treatment sample with respect to that in the corresponding control sample (see Supplementary Data 13 for replicate correlations of sgRNA abundance in control and treatment samples for Cas9 and Cas12a screens). Let us call $C_1$ and $T_1$ the control and treatment samples, respectively, for determining cutting scores. The cutting score $CS_i$ of sgRNA $i$ is defined as follows

$$CS_i = -\log_2\left(\frac{\underline{x}_{T_1,i}}{\underline{x}_{C_1,i}}\right) \quad (1)$$

where $\underline{x}_{C_1,i}$ and $\underline{x}_{T_1,i}$ indicate the total normalized read counts of sgRNA $i$ in samples $C_1$ and $T_1$, respectively, averaged across all replicates in their respective samples. A pseudocount of one is added to each raw count before normalization to prevent division by zero.

Similarly, let us call $C_2$ and $T_2$ control and treatment samples, respectively, for the estimation of the fitness score. The fitness score $FS_i$ of sgRNA $i$ is defined as follows

$$FS_i = \log_2\left(\frac{\underline{x}_{T_2,i}}{\underline{x}_{C_2,i}}\right) \quad (2)$$

where $\underline{x}_{C_2,i}$ and $\underline{x}_{T_2,i}$ are average total normalized read counts in samples $C_2$ and $T_2$, respectively, for sgRNA $i$. $FS_i$ represents the change in fitness when a gene targeted by sgRNA $i$ is knocked out.

Given a CS-threshold $T$, acCRISPR creates a *CS-corrected library* by removing any sgRNA from the original library that has a cutting score less than $T$. However, if no sgRNA for a given gene has a CS that exceeds $T$, the sgRNA with the highest CS that targets that gene is kept in the CS-corrected library.

The fitness score $FS_g$ for a gene $g$ is calculated as the average of fitness scores of all sgRNA targeting gene $g$, as follows

$$FS_g = \frac{\sum_{i \in g} FS_i}{m_g} \quad (3)$$

where $m_g$ represents the total number of sgRNA targeting gene $g$ in the CS-corrected library. $FS_g$ indicates the overall change in fitness in a particular screening condition when gene $g$ is knocked out. Since the knockout of an essential gene reduces cell fitness, essential genes would have lower fitness scores compared to non-essential genes.

acCRISPR identifies essential genes from a screening dataset by first creating a null distribution and then computing a p-value. The null distribution is assumed to be Gaussian with mean μ and standard deviation σ. This distribution represents the population of fitness scores of non-essential genes. Previous studies on essential gene identification in different yeasts have found ~20% of genes in the yeast genome to be typically essential for growth[19–21]. In addition, studies in mammalian cells have identified ~20% or fewer genes as essential for survival of various cell lines of interest[40–43]. Thus we hypothesize that genes having FS values higher than the 20th percentile in the screening dataset are putatively non-essential. The value of μ is assumed to be equal to the median of all gene FS values and σ is computed as follows:

(i)  1000 putatively non-essential genes are randomly sampled and sgRNA targeting these genes are pooled together to form an 'sgRNA pool.'

(ii)  A set of $N$ sgRNA are randomly sampled from this pool and assumed to target a pseudogene, the FS of this pseudogene is calculated as the average fitness score of the sampled sgRNA. This step is repeated to generate a total of 1000 pseudogenes.

(iii)  The standard deviation of the fitness scores of these 1000 pseudogenes is computed.

(iv)  Steps (i)-(iii) are repeated 50 times and σ of the null distribution is calculated as the average of the 50 standard deviations (obtained in step (iii)).

(v)  In these calculations, the value of $N$ is initialized to the average coverage of the original library rounded off to the nearest integer. If the total number of sgRNA to be sampled from the sgRNA pool (using this value of N) is more than twice the pool size, $N$ is reduced until this value drops below 2.

To identify essential genes, the resulting null distribution is used to perform a one-tailed z-test of significance for every gene in the dataset to determine whether its fitness score is significantly lower than μ. The raw *p*-values from the z-test are adjusted for multiple comparisons by FDR-correction and genes having corrected p-values less than a certain threshold (default: 0.05) are deemed as essential. Since every CS-threshold would result in a different essential gene set, the final set of essential genes is decided based on the value of a metric called the 'ac-coefficient', which is defined as:

$$ac - coefficient = (CS - threshold) * (avg.\,coverage\,of\,CS\,corrected\,library) \quad (4)$$

The CS-threshold at which the ac-coefficient is maximum is considered optimum, and the set of essential genes obtained at this threshold is taken as the

final essential gene set. In order to find the maximum ac-coefficient amongst values at different CS-thresholds, only those thresholds should be considered at which the average coverage of the library is >2, since a genome coverage of <2 would reduce statistical power to accurately determine gene essentiality.

acCRISPR also has the ability to analyze CRISPR screening data to identify genes that result in both positive or negative fitness effects. In this case, the fraction of genes directly related to the phenotype is typically less than the number of essential genes. Thus, we assume that 95% of genes in the screening dataset (i.e., FS values between the 2.5th percentile and 97.5th percentile) are putatively non-significant, and use them for calculating the null distribution parameters ($\mu$ and $\sigma$). Further, acCRISPR uses a two-tailed test of significance to identify hits.

**Implementation of acCRISPR with different input datasets.** acCRISPR takes raw sgRNA counts from genome-wide screens as input and processes them to calculate CS and FS per sgRNA, as described in the previous section. However, if CS and FS values have already been calculated previously or are readily available, they can be directly provided as input by skipping $log_2$-fold change calculation from raw counts.

For the CRISPR-Cas9 and -Cas12a datasets, acCRISPR was first implemented using raw sgRNA counts for all targeting sgRNA in the libraries. In subsequent acCRISPR runs, CS and FS values from the first run were input to the method (i.e., $log_2$-fold change calculation was skipped) along with a CS-threshold to identify essential genes using a CS-corrected library. For essential gene identification, a one-tailed test of significance was performed.

For implementing acCRISPR using guide activity scores from prediction algorithms, the predicted activity of each guide was provided in place of an experimentally derived CS value along with FS as input for each run. Guide activity and CS thresholds used for analyzing datasets can be found in Supplementary Table 1.

For the salt tolerance datasets, raw sgRNA counts from the control and treatment samples were used to calculate TS for each sgRNA (in the same manner as FS calculation) in the specific screening condition. These sgRNA TS values were used as input to acCRISPR in conjunction with the already calculated CS values from the essential gene analysis. Before implementing acCRISPR, sgRNA having very low normalized abundance (<2.5% of the mean normalized abundance) in the control sample for TS calculation were discarded from the library. Significant genes from acCRISPR were then determined by performing a one-tailed test of significance. In all cases, genes having FDR-corrected $p$-value < 0.05 were considered as significant.

**Implementation of other CRISPR screen analysis methods.** For implementing JACKS[10] and CRISPhieRmix[18], PO1f and PO1f Cas9/Cas12a strains of *Y. lipolytica* were used as control and treatment samples respectively.

Raw sgRNA counts from these two strains were provided as input to JACKS v0.2. To obtain p-values from JACKS, 500 genes classified as 'non-essential' by the transposon analysis[19] were randomly sampled and provided separately as negative control genes for the CRISPR-Cas9 and -Cas12a datasets. The raw $p$-values were FDR-adjusted and genes having a corrected $p$-value < 0.05 were deemed as essential.

Raw sgRNA counts from untransformed library samples were used as control (initial sgRNA abundance) and those from PO1f Cas9/Cas12a were used as treatment for MAGeCK-VISPR v0.5.6[11]. Since the data being analyzed came from negative selection screens, two-tailed raw $p$-values from Wald test were converted to one-tailed $p$-values, followed by FDR-correction. Genes having FDR-adjusted $p$-value < 0.05 were considered as essential.

CRISPhieRmix v1.1 was implemented using R 4.0.2 (Rstudio 1.4.1106) by providing $log_2$-fold changes of all sgRNA as input. The $log_2$-fold changes were calculated in a manner similar to that of fitness scores. $Log_2$-fold changes of non-targeting sgRNA in the respective libraries were provided as negative controls. The parameter *screenType* was set to 'LOF' since the sgRNA $log_2$-fold changes were obtained from negative selection screens. Genes having FDR-adjusted (1 – *genePosteriors*) values <0.05 were deemed as essential.

**Microbial strains and culturing.** All strains used in this work are presented in Supplementary Table 2. We describe the parent *Yarrowia* strain used for molecular cloning, and the related culture conditions here.

*Yarrowia lipolytica* PO1f (MatA, *leu2-270*, *ura3-302*, *xpr2-322*, *axp-2*) is the parent for all mutants used in this work. Cas9 and Cas12a expressing strains were constructed by integrating UAS1B8-TEF(136)-Cas9-CYCt and UAS1B8-TEF(136)-LbCpf1-CYCt expression cassettes into the A08 locus[9,44]. The PO1f Cas9 *ku70* and PO1f Cas12a *ku70* strains were constructed by disrupting *KU70* using CRISPR-Cas9 as previously described[17].

Yeast culturing was conducted at 30 °C in 14 mL polypropylene tubes or 250 mL baffled flasks as noted, at 225 RPM. Under non-selective conditions, *Y. lipolytica* was grown in YPD (1% Bacto yeast extract, 2% Bacto peptone, 2% glucose). Cells transformed with sgRNA-expressing plasmids were selected for in synthetic defined media deficient in leucine (SD-leu; 0.67% Difco yeast nitrogen base without amino acids, 0.069% CSM-leu (Sunrise Science, San Diego, CA), and 2% glucose). CRISPR screens for determining tolerance to high salinity were done

in SD-leu containing a final concentration of 0.75 M and 1.5 M sodium chloride. The desired salinity was achieved by the addition of an appropriate quantity of autoclaved 5 M sodium chloride stock solution.

All plasmid construction and propagation were conducted in *Escherichia coli* TOP10. Cultures were conducted in Luria-Bertani (LB) broth with 100 mg L$^{-1}$ ampicillin at 37 °C in 14 mL polypropylene tubes, at 225 RPM. Plasmids were isolated from *E. coli* cultures using the Zymo Research Plasmid Miniprep Kit.

**Plasmid construction.** All plasmids and primers used in this work are listed in Supplementary Tables 3 and 4. The plasmids used to construct Cas9 and Cas12a expressing strains of *Y. lipolytica* PO1f and the sgRNA expression plasmids were previously reported (see refs. [9,16]). We describe the construction of these plasmids again here to provide a complete accounting of this work.

For *CAS9* integration, we constructed the vector pHR_A08_Cas9, which integrates a UAS1B8-Cas9 expression cassette into the A08 locus of *Y. lipolytica* PO1f. First, pHR_A08_hrGFP (Addgene #84615) was digested with BssHII and NheI, and *CAS9* was inserted via Gibson Assembly after PCR via Cr_1250 and Cr_1254 from pCRISPRyl (Addgene #70007). Integration was accomplished as previously described using a two plasmid CRISPR-mediated markerless approach[44]. The creation of the Cas9 genome-wide library expression plasmid was facilitated by removing the Cas9-containing fragment from pCRISPRyl using restriction enzymes BamHI and HindIII, and circularizing. The M13 forward primer was used to ensure correct assembly of the construct.

*LbCAS12a* integration was accomplished in a similar manner. We first constructed pHR_A08_LbCas12a by digesting pHR_A08_hrGFP (Addgene #84615) with BssHII and NheI, and the LbCAS12a fragment was inserted using the New England BioLabs (NEB) NEBuilder® HiFi DNA Assembly Master Mix. The *LbCAS12a* gene fragment was amplified along with the necessary overlaps by PCR using Cpf1-Int-F and Cpf1-Int-R primers from pLbCas12ayl. Successful cloning of the LbCas12a fragment was confirmed with sequencing primers A08-Seq-F, A08-Seq-R, Tef-Seq-F, Lb1-R, Lb2-F, Lb3-F, Lb4-F, and Lb5-F. To create the Cas12a sgRNA genome-wide library expression plasmid (pLbCas12ayl-GW) the UAS1B8-TEF- LbCas12a-CYC1 fragment was removed from pLbCas12ayl with the use of XmaI and HindIII restriction enzymes. Subsequently, the primers BRIDGE-F and BRIDGE-R were used to circularize the vector, and the M13 forward primer was used to ensure correct assembly of the construct.

The gRNAs library vector was constructed using pCas9yl-GW (SCR1'-tRNA-AvrII site) as the backbone. The library was generated by digesting pCRISPRyl with BamHI and HindIII and circularizing to remove the Cas9 gene and its promoter and terminator using (NEBuilder® HiFi DNA Assembly). The methods used to create the guide library are provided below in the sgRNA library cloning subsection.

The LbCas12a sgRNA expression plasmid (pLbCas12ayl) was similarly constructed, but a second direct repeat sequence at the 5' of the polyT terminator in pCpf1_yl (see ref. [16]) was added. This was done to ensure that library sgRNAs could end in one or more thymine residues without being construed as part of the terminator. To make this mutation, pCpf1_yl was first linearized by digestion with SpeI. Subsequently, primers ExtraDR-F and ExtraDR-R were annealed and this double-stranded fragment was used to circularize the vector (NEBuilder® HiFi DNA Assembly).

**sgRNA library design.** sgRNA library design for the Cas9 and Cas12a CRISPR systems was accomplished as previously described in refs. [9,16]. The critical elements of the design are described again here.

Using the annotated genome of PO1f's parent strain (CLIB89; [https://www.ncbi.nlm.nih.gov/assembly/GCA_001761485.1][45]) as a reference, custom MATLAB scripts were used to design up to 8 unique Cas12a sgRNAs per gene. First, a list of all sgRNAs (25 nucleotides in length) with a TTTV (V = A/G/C) PAM were identified in both the top and bottom strand of each CDS (List A). A second list containing all possible 25nt sgRNAs with a TTTN (N = any nucleotide) PAM from the top and bottom strands of all 6 chromosomes in *Y. lipolytica* was also generated and used as a reference set to test for sgRNA uniqueness (List B). The uniqueness test was carried out by comparing the first 14nt of each sgRNA (seed sequence) in List A to the first 14nt of every sgRNA in List B. Any sequence that occurred more than once was deemed as not-unique and was removed from List A. sgRNAs that passed the uniqueness test were then picked in an unbiased manner, with even representation from the top and bottom strands when possible, starting from the 5' end of the CDS. When possible eight unique sgRNAs were selected for each gene. In cases where eight unique guides were not available, all unique guides were selected. In addition to the gene targeting guides, 651 non-targeting control guides were also designed. Random 25nt sequences were generated and each sequence was queried against the PO1f genome. Only sgRNA sequences in which the first 10nt were not found anywhere in the genome were selected and used as part of the control set.

The Cas9 sgRNA library was similarly designed, with the following differences. Working with the annotated CLIB89 genome, custom MATLAB scripts were used to identify unique sgRNAs (NGG PAM + 12 bp closest to the PAM) located within the first 300 bp of the gene. Subsequently, the top 6 sgRNAs from this filtered list were ranked based on their on-target activity score (Designer v1[25]) and the top 6 guides were selected. 480 sgRNAs with random sequence were also added to the

library as non-targeting controls. These guides were confirmed not to target anywhere within the genome by ensuring that the first 12 nt of the sgRNA did not map to any genomic locus[9].

**sgRNA library cloning**. The Cas12a library targeting the protein-coding genes in PO1f was ordered as an oligonucleotide pool from Agilent Technologies Inc. and cloned in-house using the Agilent SureVector CRISPR Library Cloning Kit (Part Number G7556A) as previously described in ref. [16].

First, the backbone pLbCas12ayl-GW was linearized and amplified by PCR using the primers InversePCR-F and InversePCR-R. To verify the completely linearized vector, we DpnI digested amplicon, purified the product with Beckman AMPure XP SPRI beads, and transformed it into *E. coli* TOP10 cells. A lack of colonies indicated a lack of contamination from the intact backbone.

Library ssDNA oligos were then amplified by PCR using the primers OLS-F and OLS-R for 15 cycles as per vendor instructions using Q5 high fidelity polymerase. The amplicons were cleaned using the AMPure XP beads prior to use in the following step. sgRNA library cloning was conducted in four replicate tubes using Agilent's SureVector CRISPR library cloning kit (Catalog #G7556A). The completed reactions were pooled and subjected to another round of cleaning.

Two amplification bottles containing 1 L of LB media and 3 g of high-grade low-gelling agarose were prepared, autoclaved, and cooled to 37 °C (Agilent, Catalog #5190-9527). Eighteen replicate transformations of the cloned library were conducted using Agilent's ElectroTen-Blue cells (Catalog #200159) via electroporation (0.2 cm cuvette, 2.5 kV, 1 pulse). Cells were recovered and with a 1 h outgrowth in SOC media at 37 °C (2% tryptone, 0.5% yeast extract, 10 mM NaCl, 2.5 mM KCl, 10 mM MgCl₂, 10 mM MgSO₄, and 20 mM glucose.) The transformed *E. coli* cells were then inoculated into two amplification bottles and grown for two days until colonies were visible in the matrix. Colonies were recovered by centrifugation and subject to a second amplification step by inoculating an 800 mL LB culture. After 4 h, the cells were collected, and the pooled plasmid library was isolated using the ZymoPURE II Plasmid Gigaprep Kit (Catalog #D4202) yielding ~2.4 mg of plasmid DNA encoding the Cas12a sgRNA library. The library was subject to a NextSeq run to test for fold coverage of individual sgRNA and skew.

The Cas9 library was constructed by the US Department of Energy's Joint Genome Institute as a deliverable of Community Science Project (CSP) 503076. Experimental details as previously described in ref. [9] are included here for completeness. The pooled sgRNA library targeting the protein-coding genes of PO1f was ordered as four oligo pools each consisting of 25% of the designed sgRNAs from Twist Bioscience and cloned. The separation into different sub-libraries was done to test different methods of assembly; the details of each approach are briefly described here.

For sub-libraries 1 and 3, second-strand synthesis reactions were conducted using the primer sgRNA-Rev2 and T4 DNA polymerase (NEB), gel extracted, and purified using Zymo Research Zymo-Spin 1 columns. For sub-libraries 2 and 4, oligos were amplified with primers via Q5 DNA polymerase (NEB) using 0.2 picomoles of DNA as a template for 7 cycles, and column purified. Library 2 had overlaps of 20 bp on either side of the spacer and was amplified with 60mer_pool-F and spacer-AarI.rev. Library 4 had overlaps of ~60 bp on either side of the spacer and was amplified with primers pLeu-mock-sgRNA.fwd and sgRNA-Rev2. Libraries 1, 3, and 4 were cloned into the AarI digested pCas9yl-GW vector using the Gibson Assembly HiFi HC 1-step Master Mix (SGI-DNA). Library 2 was digested with AarI and cloned into pCas9yl-GW digested with AarI using Golden Gate assembly with T4 DNA ligase (NEB).

The cloning method for library 4 resulted in the least number of spacers missing in the propagated library. Cloned DNA was transformed into NEB 10-beta *E. coli* and plated. Sufficient electroporations were performed for each library to yield a > 10-fold excess in colonies for the number of library variants. The plasmid library was isolated from the transformed cells after a short outgrowth.

**Yeast transformation and screening**. Transformation of the Cas9 and Cas12a sgRNA plasmid libraries into *Y. lipolytica* was done using a method previously described in refs. [9,16]. For Cas12a experiments, 3 mL of YPD was inoculated with a single colony of the strain of interest and grown in a 14 mL tube at 30 °C with shaking at 200 RPM for 22–24 h (final OD ~ 30). Cells were pelleted by centrifugation (6300 × g), washed with 1.2 mL of transformation buffer (0.1 M LiAc, 10 mM Tris (pH = 8.0), 1 mM EDTA), pelleted again by centrifugation, and resuspended in 1.2 mL of transformation buffer. To these resuspended cells, 36 μL of ssDNA mix (8 mg/mL Salmon Sperm DNA, 10 mM Tris (pH = 8.0), 1 mM EDTA), 180 μL of β-mercaptoethanol mix (5% β-mercaptoethanol, 95% triacetin), and 8 μg of plasmid library DNA were added, mixed via pipetting, and incubated for 30 mins. at room temperature. After incubation, 1800 μL of PEG mix (70% w/v PEG (3350 MW)) was added and mixed via pipetting, and the mixture was incubated at room temperature for an additional 30 min. Cells were then heat shocked for 25 min at 37 °C, washed with 25 mL of sterile Milli-Q H₂O, and used to inoculate 50 mL of SD-leu media. Dilutions of the transformation (0.01% and 0.001%) were plated on solid SD-leu media to calculate transformation efficiency. Three biological replicates of each transformation were performed for each condition. Transformation efficiency for each replicate from the Cas9 and Cas12a experiments is presented in Supplementary Table 5.

Transformation for the Cas9 library was done in a very similar manner. Briefly, half the amount of cells, DNA, and other chemical reagents described above were used for a single transformation and multiple transformations were done and pooled as necessary to ensure adequate diversity to maintain library representation and minimize the effect of plasmid instability (100x coverage, 5 × 10⁶ total transformants per biological replicate).

Screening experiments were conducted in 25 mL of liquid media in a 250 mL baffled flask (220 RPM shaking, 30 °C). Cells first reached confluency after two days of growth (OD₆₀₀ ~ 12), at which time 200 μL, which includes a sufficient number of cells for ~500-fold library coverage, was used to inoculate 25 mL of fresh media. The cells were again subcultured upon reaching confluency after four days of culture, and the experiment was stopped after reaching confluency again on day six of the screen. Glycerol stocks of day 2 cultures were also prepared and used to start other growth screens as discussed in a following subsection.

On days two, four, and six, 1 mL of culture was removed to isolate sgRNA expression plasmids for deep sequencing. Each sample was first treated with DNase I (New England Biolabs; 2 μL and 25 μL of DNaseI buffer) for 1 h at 30 °C to remove any extracellular plasmid DNA. Cells were then isolated by centrifugation at 4500 × g, and the resulting cell pellets were stored at −80 °C prior to sequencing.

***Y. lipolytica* salt tolerance screens**. CRISPR-Cas9 growth screens with high salinity were conducted in synthetic defined media deficient in leucine. Media were prepared with two different salt concentrations as defined in the microbial strains and culturing subsection. 150 μL (~1 × 10⁷ cells) of Day 2 glycerol stocks of PO1f Cas9 strain transformed with the sgRNA library were used to inoculate 250 mL baffled flasks containing 25 mL of three different media: SD-leu, SD-leu (0.75 M NaCl), and SD-leu (1.5 M NaCl). Three biological replicates were cultured for each different media condition. Outgrowth following inoculation was done at 30 °C at 225 RPM. Cells were grown for two days, and fresh media was inoculated with at least 1 × 10⁷ cells and grown for another two days. The experiment was halted after 4 days of outgrowth following inoculation. On the last day, 1 mL of culture was removed, treated with DNase I, pelleted, and processed to extract plasmids as described above. Extracted plasmids were quantified by qPCR, and amplified with forward (Cr1665-Cr1668) and reverse primers (Cr1669-Cr1671, Cr1673, and Cr1709) containing the necessary barcodes and adapters for NGS using NextSeq. Growth of the PO1f Cas9 strain in SD-leu was used as a control in the salt tolerance screens to select for genetic perturbations that conferred a growth disadvantage only under the stressed condition.

**Library isolation and sequencing**. Frozen culture samples from pooled CRISPR screens were thawed and resuspended in 400 μL sterile, Milli-Q H₂O. Each cell suspension was split into two, 200 μL samples. Plasmids were isolated from each sample using a Zymo Yeast Plasmid Miniprep Kit (Zymo Research). Splitting into separate samples here was done to accommodate the capacity of the Yeast Mini-prep Kit, specifically to ensure complete lysis of cells using Zymolyase and lysis buffer. This step is critical in ensuring sufficient plasmid recovery and library coverage for downstream sequencing. The split samples from a single pellet were pooled, and the plasmid copy number was quantified using quantitative PCR with qPCR-GW-F and qPCR-GW-R and SsoAdvanced Universal SYBR Green Super-mix (Biorad). Each pooled sample was confirmed to contain at least 10⁷ plasmids so that sufficient coverage of the sgRNA library is ensured.

To prepare samples from the Cas12a screen for next-generation sequencing, isolated plasmids were subjected to PCR using forward (ILU1-F, ILU2-F, ILU3-F, ILU4-F) and reverse primers (ILU(1–12)-R) containing all necessary barcodes and adapters for next-generation sequencing using the Illumina platform (Supplementary Table 6). Schematics of the amplicons from the Cas9 and Cas12a screens submitted for NGS are depicted in Supplementary Fig. 9. At least 0.2 ng of plasmids (~3 × 10⁷ plasmid molecules) were used as template for PCR and amplified for 16 cycles and not allowed to proceed to completion to avoid amplification bias. PCR product was purified using SPRI beads and tested on the bioanalyzer to ensure the correct length.

Samples from the Cas9 screens were prepared as previously described in ref. [9] Briefly, isolated plasmids were amplified using forward (Cr1665-Cr1668) and reverse primers (Cr1669-Cr1673; Cr1709-1711) containing the necessary barcodes, pseudo-barcodes, and adapters (Supplementary Table 7). Approximately 1 × 10⁷ plasmids were used as a template and amplified for 22 cycles, not allowing the reaction to proceed to completion. Amplicons at 250 bp were then gel extracted and tested on the bioanalyzer to ensure correct length. Samples were pooled in equimolar amounts and submitted for sequencing on a NextSeq 500 at the UCR IIGB core facility.

**Generating sgRNA read counts from raw reads**. Next-generation sequencing raw fastq files were processed using the Galaxy platform[46]. Read quality was assessed using FastQC v0.11.8., demultiplexed using Cutadapt v1.16.6, and truncated to only contain the sgRNA using Trimmomatic v0.38. Custom MATLAB scripts were written to determine counts for each sgRNA in the library using Bowtie alignment (Bowtie2 v2..4.2; inexact matching) and naïve exact matching (NEM). The final count for each sgRNA was taken as the maximum of

the two methods. A large majority of data points were derived from inexact matching with Bowtie, in only a few cases where Bowtie failed to give proper alignment, was the exact matching value used. Parameters used for each of the tools used on Galaxy for Cas12a and Cas9 screens are provided in Supplementary Tables 8 and 9 respectively. MATLAB scripts are provided as part of the GitHub link found below in the "Code availability" section. Supplementary Data 14 provides further information correlating the NCBI SRA file names to the information needed for demultiplexing the readsets. Analysis of raw Cas9 and Cas12a libraries revealed 721 and 12 sgRNA, respectively, that were found to be either missing or having very low normalized abundance (<5% of the normalized mean abundance of the library) and were discarded from further analysis (see Supplementary Data 15 for raw sgRNA counts of the untransformed Cas9 and Cas12a libraries).

**Gene ontology enrichment analysis**. GO annotations for the CLIB89 reference genome of *Y. lipolytica*[47] were obtained from MycoCosm (mycocosm.jgi.doe.gov). GO analysis for the essential gene sets was performed using the Galaxy platform[46]. First, GO-slim annotations for CLIB89 were obtained using GOSlimmer v1.0.1. Next, the GO annotation and GO-slim annotation files were used to perform GO enrichment and GO-slim enrichment analyses respectively, using GOEnrichment v2.0.1. For this analysis, the list of essential genes from a particular dataset was provided as the study set, and the list of all genes covered by the corresponding library was provided as the population set. GO terms/GO-slim terms having FDR-corrected *p*-value < 0.05 from the hypergeometric test were considered to be over-represented.

**Finding essential gene homologs in *S. cerevisiae* and *S. pombe***. Sequences of essential genes in the *Y. lipolytica* consensus set from the CLIB89 strain were aligned to genes in *S. cerevisiae* and *S. pombe* using BLASTP. *S. cerevisiae* essential genes (phenotype:inviable) were retrieved from the Saccharomyces Genome Database (SGD), and *S. pombe* essential genes were taken from Kim et al.[21]. Pairs of query and subject sequences having >40% identity from BLASTP were deemed as homologs.

**Experimental validation of essential genes and salt tolerance genes**. Selected hits from the essential gene and salt tolerance screens were validated by performing single gene knockouts using CRISPR-Cas9 genome editing and measuring the growth of these knockouts. Gene knockouts were made by using high-activity sgRNAs (i.e., sgRNA with cutting scores >5.0; see Supplementary Table 10 for a complete list). For construction of sgRNA expression vector, pCas9yl-GW was digested with AvrII, similar to the construction of sgRNA library plasmids. Primers for sgRNA cloning were obtained from Integrated DNA Technology (IDT). Each primer contained 20 bp of homology flanking either side of a 20 bp target sequence. A mixture of two primers was placed in a thermocycler to anneal the oligos together and create double stranded DNA. Next, the annealed oligonucleotide was inserted by HiFi DNA Assembly (New England BioLabs, NEB) into a linearized pCas9yl-GW vector. Successful cloning of the sgRNA fragment was confirmed by Sanger sequencing.

Cells containing integrated Cas9 were grown in YPD before being subjected to transformation of plasmid containing an sgRNA. All transformants were then inoculated in 17 × 100 mm round-bottomed polystyrene tubes containing 3 mL of SD-Leu media and allowed to grow for 16 h at 30 °C and 200 rpm shaking. Cells were then subcultured in 2 mL of fresh media with a starting $OD_{600}$ of 0.025. After 2 days of growth, cell density was determined by measuring $OD_{600}$ using a Nanodrop 2000c (Fisher Scientific) and a 1 cm pathlength cuvette. In the case of essential genes, a culture containing cells with an empty vector was used as a positive control, while the wildtype strain containing no plasmid was used as a negative control. Two biological replicates were performed for each sample.

Validation of salt tolerance genes was performed using high salinity media (SD-Leu containing 1.5 M NaCl). Cas9 expressing cells were transformed with plasmid containing sgRNA and transformants were grown in SD-Leu for 16 h. This was followed by inoculation in 2 mL of high salinity media to an initial $OD_{600}$ of 0.025. Inoculation in SD-Leu devoid of salt was used as a reference condition. After 4 days of growth in the presence and absence of salt stress, cell density was determined by measuring the $OD_{600}$. Sample containing cells with an empty plasmid was used as a positive control. Two biological replicates were performed for each sample.

**Implementation of sgRNA activity prediction tools**. DeepGuide predicted CS values for CRISPR-Cas9 and -Cas12a datasets were obtained using DeepGuide v1.0.0[16]. sgRNA activity prediction scores from Designer v1[25], Designer v2[26], CRISPRspec[29], CRISPRscan[28], SSC[27], and uCRISPR[24] were obtained using CHOPCHOP v3[48]. Similarly, DeepCpf1 scores were obtained using DeepCpf1[30].

**Calculation of sensitivity and precision**. Sensitivity measures the fraction of the consensus set of essential genes that is covered by predicted essential genes from a given method and is computed as:

$$\% \, Sensitivity = \left( \frac{No.\,of\,predicted\,essential\,genes\,overlapping\,with\,the\,consensus\,set}{Size\,of\,the\,consensus\,set} \right) * 100 \tag{5}$$

Precision measures the fraction of predicted essential genes from a given method that overlap with the consensus set and is calculated as:

$$\% \, Precision = \left( \frac{No.\,of\,predicted\,essential\,genes\,overlapping\,with\,the\,consensus\,set}{Total\,no.\,of\,predicted\,essential\,genes} \right) * 100 \tag{6}$$

**Statistics and reproducibility**. All statistical analyses performed in this study are described in the relevant Methods subsections.

**Reporting summary**. Further information on research design is available in the Nature Portfolio Reporting Summary linked to this article.

## Data availability
The sgRNA sequencing data for all CRISPR-Cas9 and -Cas12a screens generated for this study have been deposited in the NCBI SRA database under accession code PRJNA857832. Source data for main figures in the study not included in Supplementary Data 1-15 is provided in Supplementary Data 16. Any remaining information can be obtained from the corresponding author upon reasonable request.

## Code availability
Source code for acCRISPR can be found at https://github.com/ianwheeldon/acCRISPR. This GitHub page includes system requirements, instructions for installation, and usage examples. Custom Matlab scripts that were used for the design of the Cas12a CRISPR library and processing of Illumina reads to generate sgRNA abundance for both Cas9 and Cas12a screens can also be found at the same link. A permanent repository of the software has been created and archived to Zenodo (https://doi.org/10.5281/zenodo.7847623[49]).

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

## Acknowledgements
This work was supported by DOE DE-SC0019093, DOE Joint Genome Institute grant CSP-503076, NSF 1706545, NSF1803630, and NSF Plants-3D 1922642.

## Author contributions
A.R., V.T., and I.W. conceived the idea, planned the experiments, and analyzed the data. A.R., C.S., and M.L. conducted the CRISPR-Cas9 growth and salt tolerance screens. A.R. conducted the CRISPR-Cas12a screens. V.T. analyzed all screens using acCRISPR and other CRISPR screen analysis methods. S.c.L. and A.T. performed validation experiments for essential and non-essential genes. S.c.L. performed experimental validation of high salt tolerance genes. V.T., A.T., A.M., and S.L. predicted the activity of CRISPR-Cas9 and -Cas12a guides and analyzed the prediction data using acCRISPR. All authors wrote and edited the manuscript.

## Competing interests
The authors declare no competing interests.
