## [Peer Review File · Communications Biology]

Reviewers' comments:

Reviewer #1 (Remarks to the Author):

CRISPR guide RNA design studies in non-conventional microbial systems is an important area for research as we move towards more and more conventional systems. The authors, building on 2 previously published papers addressing genome scale gRNA libraries in *Yarrowia lipolytica*, create a method termed "activity correction" CRISPR (acCRISPR) employing a genome wide pooled gRNA library in a ku70- background (NHEJ-) where the double strand break will most likely lead to cell death with efficient gRNA targeting and CRISPR cutting, while less effective gRNAs will remain. Fitness studies are also performed under varying conditions to further evaluate fitness at the gene level in these libraries. The work is important, but as its a step forward from previous methods, but the authors clearly improved the overall manuscript and are moving towards validation and expansion into other species, and make it more clear the advancements this manuscript provides for studies in this area. I therefore support its publication in *Communications Biology*

Reviewer #2 (Remarks to the Author):

The manuscript describes an experimental and analytical approach for high throughput CRISPR screening to identify essential genes within non-conventional organisms. I find this work to be intriguing and of great interest to a growing number of bioengineering-based fields. The study focuses on a single organism, *Yarrowia lipolytica*, undergoing several screening conditions and the comparison of those conditions to identify cutting efficiencies, gene fitness, and gene tolerances. A designed CRISPR screening library is applied to the studied organism (and genetic variants) over the span of a few days and the harvested genetic material is analyzed through NGS methods to gather read counts of remaining sgRNAs. These read counts are compared from the different conditions / genetic variants and analyzed to infer sgRNA cutting efficiency, gene fitness impact, or tolerance. The authors provide an analysis method and software, they mention a key part of the publication. The authors directly compare their method to other approaches taken in the field for assessing essential genes within the studied organism and show a solid improvement that is consistent with previous findings in conventional yeast. Additionally, the authors test for gene tolerance under different growing conditions and identify genes suggestive of being critical to situational growth. While this manuscript and study are interesting and suggestive of a straight forward approach to identifying essential genes in non-conventional organisms, I cannot recommend it for publication at this point. I believe it would be greatly improved if the authors provided confirmatory experimentation of genetic findings and further developed their software.

Validation of findings is essential to screening assays to give support to the conclusions, and while it cannot be done at the same throughput as screening, individual testing and further characterization of key results should be expected for scientific publications. Here the authors support their findings through referencing previous studies, but provide no experimental evidence that their identified essential genes impair the fitness of the organism beyond their screening assay. For instance in the high salt screen, two genes were noted to have impacted tolerance negatively (and an additional gene positively) in Figure 5B. Using the sgRNAs that target these genes from the library (or an orthogonal genetic approach), could the authors confirm that LOF of these genes in *Y. lipolytica* lead to consistent results with their screen under the same conditions? Another approach to confirming consistency with their method could be to perform their screening approach on a well-studied organism, such as *S. cerevisiae*, in addition to informatically comparing findings cross organisms. While this latter suggestion doesn't replace the former, it would suggest the level of accuracy in the approach to identifying essential genes. This accuracy then may be translatable to non-conventional organisms, suggesting how confident findings may or may not be.

A major point of the manuscript is the presentation of their software, which while meeting the standards of the journal for supportive code for the study, falls a short of expectations in the field for informatic pipeline software for distribution. The documentation on the use of the software is minimal, there is a lack of scripted testing (unit or functional), and while GitHub is a platform for software development, it is not geared toward software persistence (such as zenodo). If the authors are attempting to develop a software that peers may be able to utilize, then more work is

warranted on their software as it will need to be held to a higher standard than reproducing a single analysis for a manuscript.

Response to Reviewers

Reviewer #1

General comment: CRISPR guide RNA design studies in non-conventional microbial systems is an important area for research as we move towards more and more conventional systems. The authors, building on 2 previously published papers addressing genome scale gRNA libraries in *Yarrowia lipolytica*, create a method termed "activity correction" CRISPR (acCRISPR) employing a genome wide pooled gRNA library in a ku70- background (NHEJ-) where the double strand break will most likely lead to cell death with efficient gRNA targeting and CRISPR cutting, while less effective gRNAs will remain. Fitness studies are also performed under varying conditions to further evaluate fitness at the gene level in these libraries. The work is important, but as its a step forward from previous methods, but the authors clearly improved the overall manuscript and are moving towards validation and expansion into other species, and make it more clear the advancements this manuscript provides for studies in this area. I therefore support its publication in Communications Biology

Response: Thank you for your review and positive feedback of our work.

Reviewer #2

General Comment. The manuscript describes an experimental and analytical approach for high throughput CRISPR screening to identify essential genes within non-conventional organisms. I find this work to be intriguing and of great interest to a growing number of bioengineering-based fields. The study focuses on a single organism, *Yarrowia lipolytica*, undergoing several screening conditions and the comparison of those conditions to identify cutting efficiencies, gene fitness, and gene tolerances. A designed CRISPR screening library is applied to the studied organism (and genetic variants) over the span of a few days and the harvested genetic material is analyzed through NGS methods to gather read counts of remaining sgRNAs. These read counts are compared from the different conditions / genetic variants and analyzed to infer sgRNA cutting efficiency, gene fitness impact, or tolerance. The authors provide an analysis method and software, they mention a key part of the publication. The authors directly compare their method to other approaches taken in the field for assessing essential genes within the studied organism and show a solid improvement that is consistent with previous findings in conventional yeast. Additionally, the authors test for gene tolerance under different growing conditions and identify genes suggestive of being critical to situational growth. While this manuscript and study are interesting and suggestive of a straight forward approach to identifying essential genes in non-conventional organisms, I cannot recommend it for publication at this point. I believe it would be greatly improved if the authors provided confirmatory experimentation of genetic findings and further developed their software.

Response: Thank you for your review of our manuscript and for your supportive comments.

Comment 1. Validation of findings is essential to screening assays to give support to the conclusions, and while it cannot be done at the same throughput as screening, individual testing and further characterization of key results should be expected for scientific publications. Here the authors support their findings through referencing previous studies, but provide no experimental evidence that their identified essential genes impair the fitness of the organism beyond their screening assay. For instance in the high salt screen, two genes were noted to have impacted tolerance negatively (and an additional gene positively) in Figure 5B. Using the sgRNAs that target these genes from the library (or an orthogonal genetic approach), could the authors confirm that LOF of these genes in *Y. lipolytica* lead to consistent results with their screen under the same conditions? Another approach to confirming consistency with their method could be to perform their screening approach on a well-studied organism, such as *S. cerevisiae*, in addition to informatically comparing findings cross organisms. While this latter suggestion doesn't replace the former, it would suggest the level of accuracy in the approach to identifying essential genes. This accuracy then may be translatable to non-conventional organisms, suggesting how confident findings may or may not be.

Response: We agree that experimental validation is a necessary part of this work. The revised manuscript contains the validation of 25 gene hits. Fifteen genes found to be essential to growth on glucose along with five non-essential genes called by acCRISPR from the CRISPR-Cas9/Cas12a were successfully validated. The validation experiments test each gene individually in conditions that mimic the screen in which they were identified. These data are reported as Supplementary Figure 2. In addition, we also validated 4 genes called in the salt tolerance screen and 1 non-significant gene from this same screen. The results of these validation experiments are now presented in Supplementary Figure 7. Details of the methods used are included in the Methods section of the manuscript. Briefly, the validation experiments sought to test whether or not the outcomes of the pooled screen could be replicated in individual knockout assays. These experiments are similar in nature to validation experiments reported for transposon data sets (see DOI: 10.1016/j.ymben.2018.05.008 and 10.1038/s41598-018-28217-z, among others).

We were not able to validate hits with high tolerance scores (TS); hits that apparently lead to improvements in growth during the stress conditions. As such, we removed the positive hit calls and edited the paper to focus on the negative TS results. The phenotype of some of the high TS hits is supported by the literature (as previously described in the original submission); however, we have elected to remove these results from the paper as validation of a select group of these was unsuccessful. We recognize that experimental validation outside of the screen context is critical and have only included screen data that is supported by validation (*i.e.*, the essential gene screen and salt tolerance screen).

Overall, 25 hits from three different screens were validated. This provides strong support that our new analysis pipeline, acCRISPR, makes accurate hit calls, which is the major conclusion of the paper. We have made edits to the abstract, introduction, results, and methods sections that described the new validation data.

Comment 2. A major point of the manuscript is the presentation of their software, which while meeting the standards of the journal for supportive code for the study, falls a short of expectations in the field for informatic pipeline software for distribution. The documentation on the use of the software is minimal, there is a lack of scripted testing (unit or functional), and while GitHub is a platform for software development, it is not geared toward software persistence (such as zenodo). If the authors are attempting to develop a software that peers may be able to utilize, then more work is warranted on their software as it will need to be held to a higher standard than reproducing a single analysis for a manuscript.

Response: We have created a permanent repository of our software on Zenodo (<https://doi.org/10.5281/zenodo.7847623>). In addition, the pH tolerance screen data now serves as an example dataset for users to test the source code on. The main conclusion of the paper – that acCRISPR calls accurate hits from CRISPR screens – is supported by two essential gene screens (Cas9 and Cas12a) and by a third screen (with two conditions) that included a high salt challenge. Hits from these screens have been validated (see response to comment 1). Based on the reviewers feedback we have also improved the documentation accompanying the software.

Updated figures

Figure 5. acCRISPR analysis of salt tolerance screens. (a) Schematic of the CRISPR-Cas9 stress tolerance screens in *Yarrowia*. Analogous to fitness score (FS), the tolerance score (TS) is used to define the effect of each guide on cell growth under a stress condition. TS is equal to the \log_2 -fold change of sgRNA abundance in the treatment to the control, where the control is a Cas9-expressing strain grown under standard culture conditions. (b) Outcomes of high salt tolerance screens. Venn diagram (top) shows the overlap of gene hits identified in the salt (0.75

M and 1.5 M NaCl) screens. Selected hits are shown (bottom), including the gene ID, the TS value from the 1.5 M NaCl condition, and putative gene function.

(Panel b modified to only include high salt tolerance screen results)

Supplementary Figure 2. Experimental validation of CRISPR-Cas9 and CRISPR-Cas12a essential and non-essential genes from acCRISPR analysis. Final OD of cells containing single knockouts of 5 non-essential genes (red bars) and 15 essential genes (green bars) from the consensus set. Of the 15 selected for validation, 12 were called as essential genes in all 3 screens (Cas9, Cas12a and transposon⁴). The other three genes, YALI1_B03043g, YALI1_E18269g and YALI1_F34105g, were called as essential only in the Cas9 and Cas12a screens. Cells were grown in SD-Leu for 16 hrs post sgRNA transformation, followed by subculturing in fresh media and growth for another 2 days before measuring final OD. An empty vector control (blue bar) was used to show growth in absence of any knockout. The PO1f strain containing no plasmid (indicated as WT; leftmost bar) was used as no growth, negative control. Each bar represents mean of two biological replicates (n = 2), data points represent OD of each individual replicate in the respective sample, and error bars represent standard deviation (****p < 0.0001 ; one-tailed unpaired t-test).

(new figure added to show essential gene validation results)

Supplementary Figure 7. Experimental validation of high salt tolerance genes from acCRISPR analysis. Bars represent final OD of (a) 5 single gene knockouts (*i.e.*, 4 significant genes and a non-significant gene, YALI1_C11819g), and (b) an empty vector control, grown in absence (normal; red bars) and presence (1.5 M NaCl; blue bars) of high salt conditions. Cells were grown in SD-Leu for 16 hrs post sgRNA transformation, followed by subculturing in fresh media (containing 1.5 M NaCl for the high salt condition) and growth for 4 days before measuring final OD. Bars indicate mean of two biological replicates (n = 2), data points represent OD of each individual replicate in the respective sample, and error bars represent standard deviation (**p < 0.01, ***p < 0.001 ; one-tailed unpaired t-test).

(new figure added to show salt tolerance screen validation results)

REVIEWERS' COMMENTS:

Reviewer #2 (Remarks to the Author):

The authors have clearly improved the overall manuscript with the addition of the validation data that further supports their conclusions. They have updated to their code base in ways that will make it easier for others to use appropriately and have submitted a copy to a persistent code repository. My comments about the manuscript have clearly been addressed and I support the publication of the manuscript in Communications Biology.

Response to Reviewers

Reviewer #2

General comment: The authors have clearly improved the overall manuscript with the addition of the validation data that further supports their conclusions. They have updated to their code base in ways that will make it easier for others to use appropriately and have submitted a copy to a persistent code repository. My comments about the manuscript have clearly been addressed and I support the publication of the manuscript in Communications Biology.

Response: Thank you for your positive feedback on our manuscript revisions.